# Environmental Impacts of Foods in the Adventist Health Study-2 Dietary Questionnaire

**Andrew Berardy [1,*,†], Ujué Fresán [1,2,3,†], Rodrigo A. Matos [4], Abigail Clarke [1], Alfredo Mejia [1], Karen Jaceldo-Siegl [1] and Joan Sabaté [1]**

1   School of Public Health, Loma Linda University, Loma Linda, CA 92318, USA; ufresans@navarra.es (U.F.); AEClarke@llu.edu (A.C.); mejiamax@gmail.com (A.M.); kjaceldo@llu.edu (K.J.-S.); jsabate@llu.edu (J.S.)
2   Centro de Investigación Biomédica en Red Epidemiología y Salud Pública (CIBER-ESP), 28029 Madrid, Spain
3   Instituto de Salud Pública y Laboral de Navarra, 31003 Pamplona, Spain
4   School of Engineering, Universidad Peruana Union, Lima 07001, Peru; amatosch@upeu.edu.pe
*   Correspondence: AndrewBerardy@llu.edu
†   First two authors contributed equally and are listed in arbitrary order.

**Abstract:** The objective of this study was to use life cycle assessment to estimate the environmental impacts (from farm to factory gate) of the 198 hard-coded line-items included in the food frequency questionnaire of the Adventist Health Study-2 survey and to assess differences among food groups. Life cycle inventories were created using existing data sources and primary data, and their global warming potential (GWP), land use, and water consumption impacts were assessed using the ReCiPe 2016 methodology. In addition to presenting the impacts according to weight and protein content across food groups, we include the novel addition of presenting impacts according to the NOVA classification indicating various levels of processing. Food categories were compared based on one kilogram of edible food, protein food sources were compared based on one kilogram of protein, and NOVA comparisons were based on one serving. In general, meats had the highest environmental impacts per both weight and protein content, while the lowest overall impacts per kilogram came from fruits. Meat analogs had the lowest overall impacts per kilogram of protein, contrary to expectations that additional processing would result in higher environmental impacts when compared to whole plant-based foods. Per serving, ultra-processed foods had the highest GWP, processed foods the highest land use, and minimally processed foods the highest water consumption. Results from this analysis were consistent with other studies. Results from this study suggest that meat and ultra-processed foods have the overall worst environmental impacts, but high water consumption in some minimally processed foods means that those should be carefully considered as well.

**Keywords:** food systems; dietary environmental impact; greenhouse gas emission; land use; water consumption

## 1. Introduction

### 1.1. Context and Importance

The consequences of food systems are a key aspect to consider for the advancement of sustainable development goals [1]. Despite the potential for food systems to improve human health and protect the environment, they are threatening both [2]. Significant environmental and human health issues can be traced back to food systems, especially Western diets [3–6]. The myriad environmental consequences of food production include deforestation, global warming, and eutrophication [7–10]. These and other impacts threaten to exceed the carrying capacity of Earth's natural systems that create a safe operating space for humanity [11]. Therefore, it is imperative that we understand the consequences of food choices.

Examining the most commonly consumed foods in the population can be accomplished through the use of a food frequency questionnaire (FFQ), which assesses how often portions of certain foods are consumed by the respondent. To have an accurate approach, the FFQ needs to be comprehensive, comprising a wide list of products, from plant-based to animal-based products, from unprocessed to highly processed foods, and from traditional products to those recently introduced in the market. An example of such a questionnaire is the Adventist Health Study-2 (AHS-2) FFQ, which consists of over 200 foods commonly consumed among Adventists [12,13]. The AHS-2 is a prospective cohort study of over 96,000 adult Adventists, geographically spread throughout North America (all 50 United States and Canadian provinces), with the goal of understanding links between diet, lifestyle, and health outcomes. Details about the establishment of the cohort have been published elsewhere [14]. The AHS-2 FFQ was applied to the AHS-2 cohort beginning in 2001 and ending in 2007. The demographic and dietary profiles of the AHS-2 cohort have been previously described [15]. Briefly, participants are relatively health-conscious, with a wide range of dietary patterns (8% vegan, 28% lacto-ovo vegetarian, 16% semi- or pesco-vegetarian, 48% nonvegetarian) and participants consume more meat analogs compared to the general population; however, the consumption of meat analogs is becoming more mainstream. The AHS-2 participants were 65% female, 65% non-Hispanic white, and 27% Black, with a mean age of 60 [14]. Despite some limited generalizability of findings from this faith-based population to the general North American population, we have no evidence of biological differences in response to environmental factors [16].

## 1.2. State of the Research Field

With origins tracing back to the 1970s and standardization of methodology beginning in the 1990s, life cycle assessment (LCA) is recommended as an environmental assessment tool for products, and widely utilized, in part, because of its integrated framework, impact assessment, and data quality [17]. Growing concern regarding the environmental impacts of food resulted in the application of the LCA methodology to evaluate food choices.

A 2018 study using Economic Input–Output LCA (EIOLCA) found that 16% of all US greenhouse gas emissions (GHGEs) came from food in 2013, with animal proteins accounting for 30% of those emissions, the highest for any single category of foods [18]. This finding is in agreement with extensive previous studies indicating that, whether based on weight, calories, or protein provided, animal-based foods are worse for the environment than plant-based foods [7,19,20].

However, partially due to the specificity required and difficulty of obtaining primary data, there is a relative lack of LCA addressing highly processed foods, which is important due to the possibility that extensive processing may increase the environmental impacts of plant-based foods to rival or exceed those of animal-based foods [3,21]. Current dietary patterns are far from the traditional minimally processed patterns, including a significantly higher proportion of highly processed food. Thus, as the food landscape evolves, new types of foods deserve increased attention to obtain an accurate estimation of the environmental impact of diets. Indeed, LCA has previously been used to effectively debunk faulty assumptions, including the notion that choosing local foods was the most important decision to make in reducing environmental impacts, by finding instead that GHGEs from food are dominated by their production phase, not their transportation [6].

The majority of LCA has a focus on GHGEs, but this is only one aspect of the potential environmental impacts caused by the production of food [18,22]. Some studies include consideration of additional impacts, such as land use or energy use [10,23]. There is a general agreement between different environmental indicators and GHGEs [24], but some tradeoffs among them could exist [25].

To better inform the consequences of dietary choices, it is important to understand these potential tradeoffs among foods that people commonly consume. To assess actual or reported consumption patterns, analysis of questionnaires can be useful. For example, the examination of the EPIC-Oxford study found GHG emissions of meat-eaters to be roughly twice as high as those of vegans based on

the application of previous analysis of food commodities [26]. Further analyses examining additional foods using multiple environmental metrics would provide a more complete approach.

*1.3. Objectives*

The goal of this research was to help fill the data gap regarding the environmental consequences of food choices based on dietary questionnaires. This was accomplished through the assessment of the GHGE, land use, and water consumption of the foods included in the AHS-2 FFQ. Differences in impacts are evaluated across food groups, between different food sources of protein, and based on the extent of processing of foods.

## 2. Materials and Methods

*2.1. Life Cycle Assessment*

LCA was chosen as the methodology to evaluate the environmental impacts of the AHS-2 FFQ foods because it facilitates detailed examination and comparison of foods, is widely used, and, therefore, has a substantial amount of data available, and encourages a comprehensive approach through life cycle thinking. Other methods to evaluate environmental impacts exist, such as carbon footprinting, material flow analysis, or economic input–output analysis [27–29]. However, these alternatives only offered a narrower scope, lack of characterization, or too broad of a scope for the goals of our analysis.

LCA is a method used to quantify the environmental impacts of goods and services, including food. The steps necessary to complete an LCA study include:

1. Definition of the LCA's goal and scope;
2. Collection of data to create a life cycle inventory of inputs and outputs;
3. Life cycle impact assessment of the data;
4. Interpretation of the assessment results.

2.1.1. Goal and Scope

The purpose of this LCA was to present the environmental impacts associated with the foods comprising the AHS-2 FFQ. These foods were assessed through a life cycle assessment utilizing primary, secondary, and tertiary data. Food categories included fruit, vegetables, meat, dairy, beverages, plant-based meat analogs, and prepared foods [14]. We utilized a comprehensive approach to include 248 foods from the extensive questionnaire with the highest degree of accuracy possible. These 248 foods were based on the 198 hard-coded items, some of which included multiple foods per item. Individual foods were grouped into twelve categories for analysis and presentation of the results. These twelve categories included *Fruits; Vegetables; Legumes; Grains, Bread, and Pasta; Breakfast Cereals and Baked Desserts; Nuts and Seeds; Dairy and Eggs; Meats, Poultry, and Fish; Meat Analogs; Mixed Dishes; Dressings, Margarine, and Oils; and Beverages.* Analysis of these categories should, therefore, provide insight into the environmental impacts associated with a wide range of foods that could be consumed by a diverse population. Individual foods comprising each of the twelve categories are listed in *Supplemental Data*.

Functional Unit

Foods serve many different purposes, which makes the selection of a functional unit difficult, especially for such a wide variety of foods [3]. This study, therefore, included a weight-based functional unit of one kilogram of edible food, which is easily translated to nutritional metrics of concern based on nutritional facts data. For the purposes of further analysis, this study also employed a functional unit of one kilogram of protein content when comparing food sources of protein, and a functional unit of one serving size when comparing foods based on their NOVA classifications. The NOVA method enables the classification of foods according to the extent of their processing, which facilitates such a comparison [30].

System Boundaries

The system boundaries for this research were from cradle to farm or manufacturer gate, depending upon the product under consideration. Therefore, agricultural production, processing, and manufacturing, where appropriate, were included. Transportation to import out-of-season fruits was included, but not for in-season products. Packaging was considered only when it was an essential component of the product, such as for carbonated beverages, and in such cases, only the primary packaging required was considered. Additional packaging for transportation or labeling was not considered. Items, such as fruits and vegetables, were assumed to have no packaging. Retail, consumption, and disposal were not included. Exact system boundaries for all foods are provided in *Supplemental Data*.

### 2.1.2. Lifecycle Inventory

Raw Ingredients Production

Data required to complete the life cycle inventory (LCI) include many ingredients due not only to items that are based on recipes with multiple ingredients (e.g., fruit salad) but also to items consisting of multiple foods on one line (e.g., one line-item included three stone fruits: peaches, nectarines, and plums). As an example, the FFQ category of "Fruits" comprised 26 hard-coded line items, which contain 31 individual foods, which require 34 different ingredients. For this reason, out-of-season fruits were counted in addition to in-season fruits when calculating impacts for the "Fruits" category. In total, the final number of foods included in the analysis was 248.

For foods with multiple ingredients, such as commercial products and mixed dishes, recipes developed by AHS-2 researchers were used to understand the constituent components of foods included in the FFQ with multiple ingredients. In some cases, the exact ingredient was not listed in the recipe, so we made assumptions regarding what it might be. For example, when a recipe called for starch or oil without specifying the type, we assumed those to be corn starch and soybean oil. Recipes for foods utilizing multiple ingredients prioritized the use of entries based on primary data and used existing secondary data from LCI databases.

Secondary data for 46 fruits and vegetables from cost studies performed by UC Davis researchers in the context of California were used to estimate the LCI for most produce items. Tertiary data from databases were used for most of the remaining LCI data. These databases included Ecoinvent 3, Agri-footprint, LCA Food DK, and Agribalyse. Additional details for data source choices are provided in the *Supplementary Information* and in *Supplemental Data* documents.

Processing

For the majority of foods that required cooking, we assumed this was done at home, and, therefore, the energy usage was outside of our system boundaries. For example, energy for cooking pizza or pasta was not included. However, the energy for processing common ingredients, such as flour, was typically included. Additional details for energy assumptions are provided in the *Supplementary Information* document in Section 1.2.

Primary data for type and amount of energy used during production were gathered onsite for 34 meat analogs at their respective factories, and the methodology for this data collection is included in an already published paper [31]. This data collection focused on factory level processing. Any data gaps for energy use for processing were filled by food science engineer and co-author, Rodrigo A. Matos. Data sources are detailed in Table S1.

Packaging and Transportation

Packaging inventory data were estimated based on the type and weight of the packaging material considered for items that include it. Data for these materials were from tertiary LCI databases, as described in "Raw ingredients production" with further details available in *Supplementary Information*.

For out-of-season produce that was assumed to be imported, we included the transportation required to deliver the produce to the United States at a major port of entry. We assumed that by land, this was to Nogales, Arizona, and by sea, this was to Los Angeles, California [32]. For transportation by sea, we assumed an oceanic freighter with cooling to Los Angeles as the mode of transit. For transportation by land, we assumed refrigerated trucks to Nogales, Arizona as the mode of transit. This transportation is intended to capture the added impacts from international shipping when compared to in-season produce that is grown within the United States. Out-of-season fruits have the added burden of transportation to the United States from their international origins and, therefore, have higher impacts than in-season berries. Therefore, transportation from the port of entry to a regional distribution center or retailer and final transportation to consumer homes was not included. Additional details for transportation assumptions are provided in Section 1.1 of *Supplementary Information*.

### 2.1.3. Life Cycle Impact Assessment

Assessment of life cycle impacts utilized SimaPro software, release 9.0.0.49. Analysis using the life cycle impact assessment (LCIA) methodology, ReCiPe 2016 Midpoint (H) V1.01 (hierarchist perspective), for 1 kg of each food was used for determining the midpoint indicators for each environmental impact category of interest. Details of the ReCiPe 2016 life cycle impact model are provided in a report describing the update from ReCiPe 2008 [33]. The ReCiPe hierarchist perspective calculates global warming potential (GWP) as the amount of additional radiative forcing over 100 years caused by the emission of 1 kg of a greenhouse gas (GHG) relative to the additional radiative forcing over 100 years caused by the emission of 1 kg of $CO_2$. This results in GWP values of 1 kg carbon dioxide equivalents ($CO_2$-eq) per kg of $CO_2$, 34 kg $CO_2$-eq per kg of methane, and 298 kg $CO_2$-eq per kg of nitrous oxide. Water consumption is defined as the amount of water the watershed of origin is losing and is considered 10% of water use for industrial and domestic applications, but 44% of water use for agriculture. So, for every 100 $m^3$ of water used in agricultural settings, 44 $m^3$ of water is consumed, while only 10 $m^3$ of water is consumed per 100 $m^3$ used in industrial and commercial settings. Land use is defined as the local land use, including transformation, occupation, and relaxation, relative to an assumed natural situation had no land use occurred. Further details regarding GWP, water consumption, and land use characterization are available in a report describing ReCiPe 2016 [33].

### 2.1.4. Interpretation of Results

Results for GWP, land use, and water consumption per kilogram were entered into Excel for subsequent analysis. To understand the average environmental impacts across different types of foods, foods were placed into common categories primarily based on food groups considered in the AHS2-FFQ, but with some additional categories for foods that did not quite fit just one, such as mixed dishes, which incorporate foods from multiple food groups. Certain items were repeated across categories. For example, several brands of margarine were included, but all with the same assumed inventory data. To avoid double-counting when considering the environmental impacts in a category of foods, such items were only included once when the inventory data were the same as other brands.

To evaluate differences in environmental impacts across different levels of processing, foods were categorized from unprocessed to ultra-processed based on the NOVA food classification system [30]. The NOVA classification is divided into four groups (1–4) from least to most processed foods. Group 1 consists of unprocessed or minimally processed foods, which are edible parts of plants or animals that may have undergone minimal processing for preservation or preparation for eating. Group 2 consists of processed culinary ingredients, such as oils, sugar, or salt, that are derived from Group 1 foods and are intended to be used with Group 1 foods to create meals. Group 3 consists of processed foods that combined Group 1 and Group 2 foods to improve the shelf life or sensory qualities of Group 1 foods. Group 4 consists of ultra-processed foods that are mostly formulations based on combinations of foods and additives, including soft drinks, packaged snacks, and pre-made frozen meals, that make use of substances extracted from foods through chemical and/or mechanical processing.

## 3. Results

### 3.1. Differences between Food Groups on a Weight Basis

The *Supplementary Information* document for this article lists the individual foods we analyzed that comprise each food group. Figure 1 shows the mean environmental impacts and standard error as an error bar for each food group.

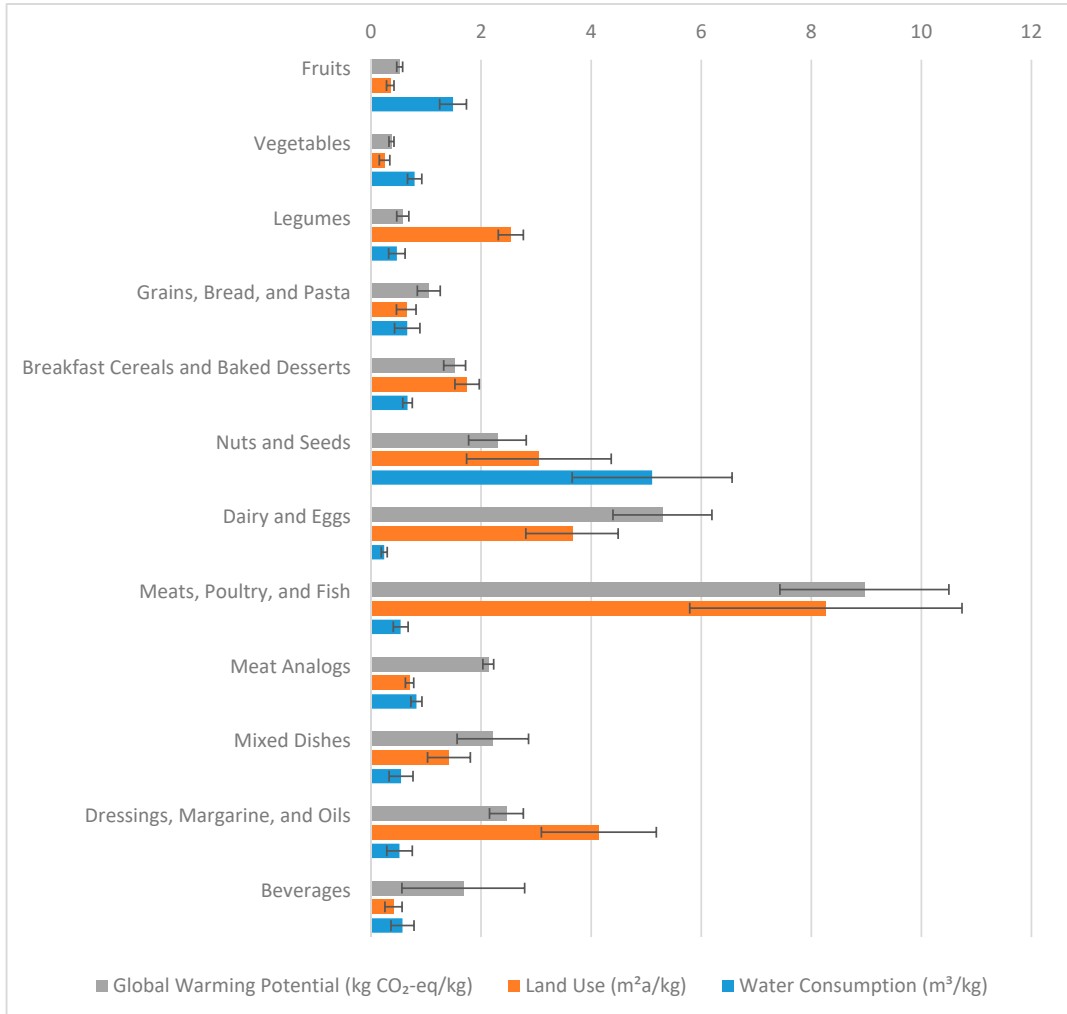

**Figure 1.** Environmental impacts by food group. Bars represent the mean environmental impact values for global warming potential, land use, and water consumption for the 12 food categories into which the AHS-2 foods were divided per kilogram of edible weight. Error bars represent the standard error for the three environmental impacts in each food category. Mixed dishes include macaroni and cheese, microwave popcorn, cheese pizza, potato salad, tuna noodle casserole, and tuna salad. Please refer to *Supplemental Data* for a complete list of foods in each category.

Per kilogram of edible food, fruits had the lowest GWP and land use impact but the second-highest water consumption. Vegetables also had fairly low GWP and land use but had lower water consumption. Grains required even less water but also took up more land. For both dairy and meats, poultry, and fish, GWP and land use were some of the highest in the list, but some of the lowest water consumption. In contrast, meat analogs had much lower GWP and land use, although they had higher water consumption. Nuts and seeds had impacts somewhere between other crops and animal protein foods for GWP and land use, but higher water consumption. Beverages were almost entirely water, so the associated environmental impacts were fairly low and depended upon the consequences of

production for the remaining ingredients, such as coffee beans, tea leaves, and sugar. Certain food groups, especially for animal-based foods, as well as nuts and seeds, and beverages, had significant variation within the group, indicating that there was a wide range of impacts across the different foods included within these groups. The second sheet in the *Supplemental Data* document presents the numerical values of the mean and standard deviation for the 12 food groups shown in Figure 1, as well as their minimum, 25th percentile, median, 75th percentile, and maximum values of each food group across the three environmental impacts.

### 3.2. Differences between Food Groups on a Protein Content Basis

Figure 2 compares the three environmental impacts for the protein from different food groups commonly consumed as sources of protein. The bars represent mean environmental impact values per kg of edible protein content from the different food sources, while the error bars show the standard error within each category. Units for the environmental impacts are kg $CO_2$ equivalents for global warming potential, $m^2a$ for land use, and $m^3$ for water consumption.

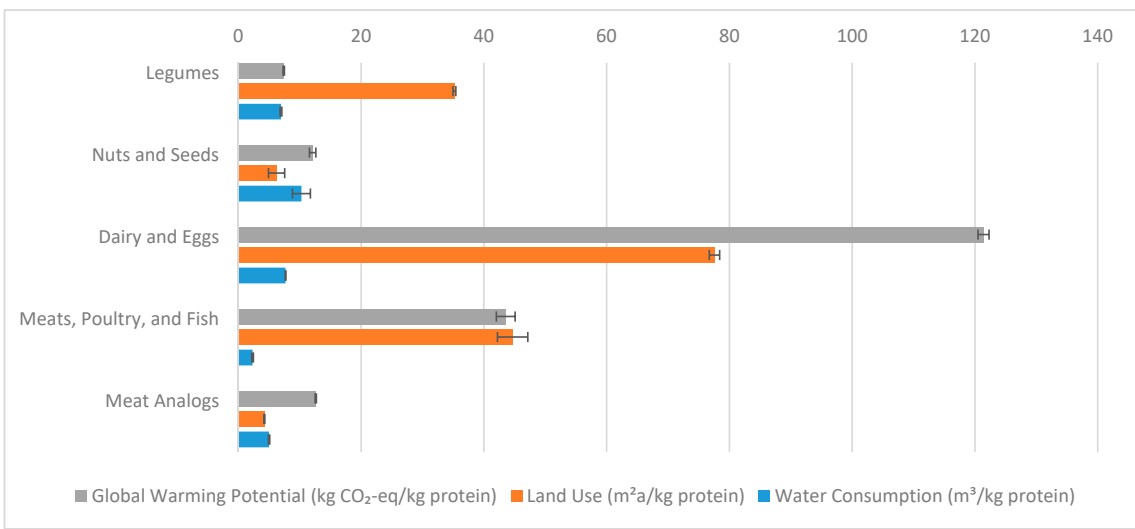

**Figure 2.** Environmental impacts of the protein derived from different food sources. Bars represent mean environmental impacts foods grouped into five categories commonly consumed for their protein content, including *Legumes; Nuts and Seeds; Dairy and Eggs; Meats, Poultry, and Fish; and Meat Analogs*. Error bars represent the standard error for the three environmental impacts in each food category. Please refer to *Supplemental Data* for a complete list of foods in each category.

As shown in Figure 2, based on one kilogram of protein content, the largest environmental impacts for GWP and land use were from dairy and eggs, followed by meats, poultry, and fish. Water consumption per kilogram of protein content was highest among nuts and seeds, followed closely by dairy and eggs.

### 3.3. Differences between Food Groups by Level of Processing

Based on NOVA classification, 42% of foods were minimally processed (group 1), 41% were ultra-processed (group 4), 13% were processed (group 3), and 4% were processed culinary ingredients (group 2). Unprocessed foods were primarily fruits, vegetables, legumes, grains, meat, and fish. Processed culinary ingredients were butter and oils. Processed foods included soups, sauces, some eggs and dairy products, and some simple mixed dishes. Ultra-processed foods included commercial cereals and breads, processed meats and meat analogs, and some beverages, including alcohol and soda. The high proportion of ultra-processed foods in the questionnaire was a result of the extensive and detailed content of 34 meat analogs and 16 commercial breakfast cereals, among other ultra-processed foods. Figure 3 presents environmental impacts per food serving across the four NOVA classifications.

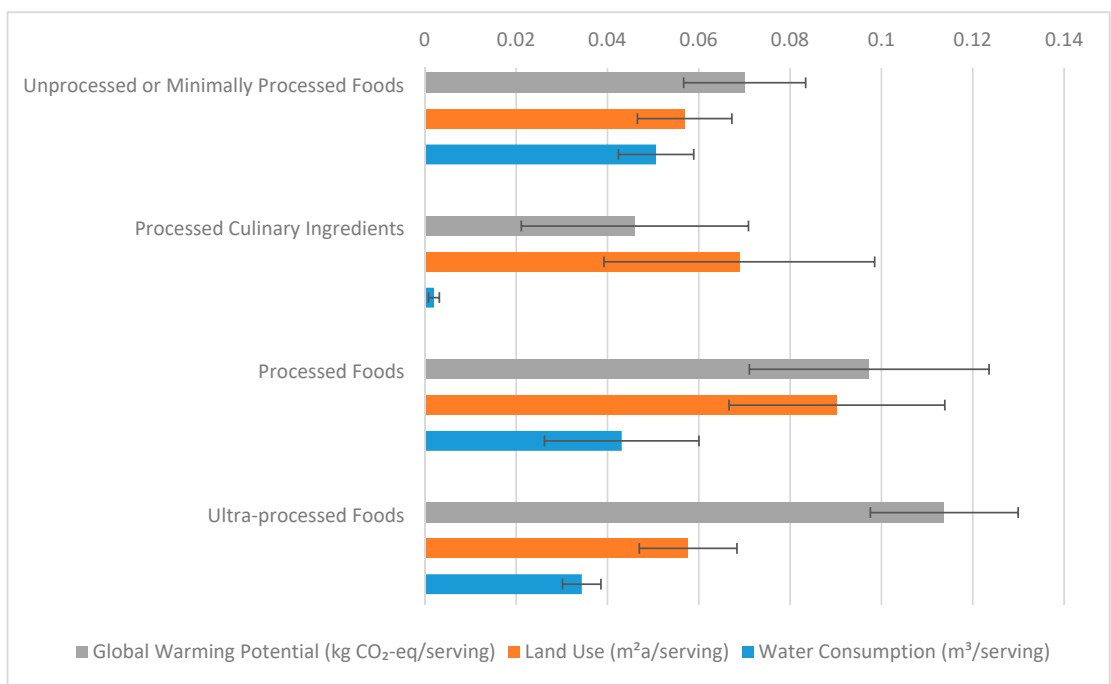

**Figure 3.** Environmental impacts per serving across NOVA classifications, which indicate the extent to which the food was processed. Bars represent the mean environmental impact values for global warming potential, land use, and water consumption per serving for four NOVA food classification categories representing the extent of processing from lowest to highest as unprocessed or minimally processed foods, culinary ingredients, processed foods, and ultra-processed foods. Error bars represent the standard error for the three environmental impacts in each food category. Please refer to *Supplemental Data* for a complete list of foods in each category.

The bars represent mean environmental impact values per serving of edible food, while the error bars show the standard error within each category. Units for the environmental impacts are kg $CO_2$ equivalents for global warming potential, $m^2a$ for land use, and $m^3$ for water consumption.

Ultra-processed foods had the highest GWP, while processed foods had the highest land use, and unprocessed and minimally processed foods had the highest water consumption. The lowest GWP was from processed culinary ingredients, while the lowest land use was from unprocessed and minimally processed foods, and the lowest water consumption was from processed culinary ingredients. There was significant variability within each NOVA classification category for all three environmental impacts, with the exception of water consumption in processed culinary ingredients. The wide variety of foods included in each category, including plant-based and animal-based foods, resulted in this large range of environmental impacts.

## 4. Discussion

The findings presented here are the result of a long and laborious process to complete an LCA of foods from an extensive and detailed questionnaire designed for the diverse Adventist population. A previous preliminary analysis of GHGEs associated with dietary patterns based on the AHS-2 FFQ was performed by Soret et al. [34]. This study expanded upon the initial assessment work of Soret et al. to consider additional environmental impacts, including land use and water consumption, as well as a more thorough analysis focused exclusively on the foods included in the AHS-2 rather than respondents' dietary patterns. Indeed, we believe these results are the most comprehensive possible based on the data we acquired. In addition to traditional food groups, we included recipes and commercial products with varying degrees of processing, as well as meat analogs. As trends move

towards people consuming more processed foods and meat analogs, understanding the impacts of these foods will have increased relevance for the general population [35,36].

This project required matching a wide range of foods with specific AHS-2 recipes. Additionally, as a result of considering many processed products, one of the challenges for the task at hand was that there were some foods for which no other LCA data were available. Thus, data about type or amount of energy used for processing raw ingredients into the final products were not available. However, we overcame this obstacle through consultation with an expert in food science. Therefore, this LCA study provided precise data on the several environmental impacts of a wide variety of specific food products.

Environmental impacts for the twelve categories of foods on a per kilogram of edible food basis provided nuanced insight into the differences for a wide variety of foods based on the category they belong to and the potential tradeoffs made between different food choices. For example, dairy and eggs had high GWP and land use impacts but low water consumption. In contrast, fruits had low GWP and land use impacts but the second-highest water consumption. Some categories were also revealed to have significant internal variation in environmental impacts, such as meats, poultry, and fish, or beverages, implying that some choices within such categories may have far higher or lower impacts than expected based on the average. Comparisons across different food sources of protein confirm the typical expectation that animal-based protein sources have higher environmental impacts overall when compared to plant-based protein sources [37,38]. This outcome was expected due to the fact that livestock must not only be fed to be productive, but also because biological processes, especially in ruminant animals, produce methane, a potent GHG. Dairy and eggs, in particular, had higher impacts even when compared to meat on this basis due to having relatively lower protein content but similar input requirements and associated emissions. Meat analogs performed very well on this basis, in part, due to the opposite reason, having high protein content. They proved to be much more efficient than animals in processing raw materials into protein. Legumes, as well as nuts and seeds, had environmental impacts that mostly fell between the extremes of the other categories. Nuts and seeds did have the highest water consumption on a protein content basis, but this is, in part, due to the small edible mass (just the seed) compared to the overall plant biomass.

NOVA classifications provided insight into how foods across different levels of processing performed in regard to their environmental impacts. Note that as these impacts are presented on a per-serving basis, processed culinary ingredients were typically used in smaller amounts than any other foods. While the expectation would be that lower levels of processing result in lower environmental impacts, the reality is that a mix of various plant- and animal-derived foods within each category resulted in no one category dominating the environmental impacts as the best or worst. Ultra-processed foods had the highest GWP as expected, but processed foods had the highest land use, and unprocessed or minimally processed foods had the highest water consumption. These findings reinforce the idea that the production of raw ingredients for a food may be a more important hotspot for its environmental impacts than the extent of its processing [39]. As a result of this generalization and the added difficulty of evaluating processing, it is often ignored, but there are some cases where it can significantly increase the overall environmental impact even if it is as simple as frozen spinach [40]. The impact of processing may be especially important when considering plant-based products, as their ingredients typically have low environmental impacts from production, therefore, increasing the proportion of environmental impacts that are derived from processing.

Accuracy and reliability of data are important concerns to address for LCA, especially when dealing with a large dataset. We compared the environmental impacts for individual foods, when available, as well as statistics for each category of foods. We took the approach of comparing to multiple other existing data sources when available, including additional LCI libraries, review studies, and individual LCA papers. As we already drew from a variety of LCI library sources, the libraries used for comparison varied based on the libraries used as the main data source. For foods with multiple ingredients,

we relied more heavily on individual peer-reviewed publications due to a lack of data in LCI libraries. Our findings are compared to those from other sources in Table 1.

**Table 1.** Mean environmental impacts calculated in this study as compared to others. For all three impacts, the first column presents the mean value found for this study and the second column presents a comparison with a previously published value or range of values for reference. For fruits, vegetables, grains, and dressings, we compared to the mean of all foods in these categories from the Agribalyse LCI database [41]. Additional details regarding the environmental impacts for each category, including their standard deviations, are available in the *Supplemental Data*. Comparison data for mixed dishes were not available.

| Food Categories | Global Warming Potential kg $CO_2$-eq per kg | | Land Use $m^2$a per kg | | Water Consumption $m^3$ per kg | |
|---|---|---|---|---|---|---|
| | This Study | Comparison [Source] | This Study | Comparison [Source] | This Study | Comparison [Source] |
| Fruits | 0.521 | 0.489 [42] | 0.349 | 1.003 [42] | 1.492 | 0.267 [42] |
| Vegetables | 0.377 | 0.292 [42] | 0.231 | 0.512 [42] | 0.815 | 0.030 [42] |
| Legumes | 0.577 | 1.226 [43] | 2.541 | 3.0 [44] | 0.470 | 0.015 [43] |
| Grains, Bread, and Pasta | 1.050 | 1.036 [42] | 0.641 | 1.763 [42] | 0.659 | 0.024 [42] |
| Breakfast Cereals and Baked Desserts | 1.520 | 2.64 [45] | 1.746 | 2.51 [45] | 0.663 | 0.743 [45] |
| Dairy and Eggs | 5.295 | 1–22 [20] | 3.652 | 1–17 [20] | 0.241 | 0.1–0.2 [46] |
| Dressings, Margarine, and Oils | 2.461 | 1.558 [42] | 4.141 | 4.601 [42] | 0.517 | 0.110 [42] |
| Meats, Poultry, and Fish | 8.966 | 1–129 [20] | 8.265 | 2–420 [20] | 0.539 | 0.003–0.221 [47] |
| Meat Analogs | 2.130 | 1–6 [20] | 0.701 | 1–3 [20] | 0.825 | 3.8 [48] |
| Nuts and Seeds | 2.297 | 2.18 [49] | 3.050 | 4.494 [50] | 5.109 | 0.175 [51] |
| Beverages | 1.677 | 0.151–0.555 [52] | 0.409 | 2.571 [42] | 0.573 | 0.376 [42] |

Out of the 33 comparisons, 22 were within 1 unit (1 kg $CO_2$-eq, 1 $m^2$a, and 1 $m^3$ per kg, for GWP, land use, and water consumption, respectively) of the comparison environmental impacts or within the range of reported environmental impacts estimated by other sources for similar categories of food, or individual foods typical of the category. Moreover, for both GWP and land use, 9 out of the 11 comparison estimates were within 1 standard deviation of our estimates. For water use estimates, only 6 were within 1 standard deviation. Standard deviations for our estimates are presented in *Supplemental Data*. Results from the current study are also consistent with other studies in their general finding that, on average, animal-based foods have higher environmental impacts than plant-based foods [6,7,19,20,26].

Our estimate of water consumption for nuts and seeds was higher than a published estimate for almonds [51], but this is reasonable given the significant variation in water consumption associated with almonds even within California alone [53]. We used a land use value for almonds that was per kilogram of protein as a comparison for our nuts and seeds category and converted it to a per kilogram of edible weight value using the assumption that it was 21.2% protein by weight [50,54]. Our water consumption estimate for meats, poultry, and fish was higher than the comparison range. It should also be noted that there is a very large range of environmental impacts associated with

animal-based foods, as reported in a review of studies, which can vary significantly based on the production practices employed [20]. Our comparison source for "Dairy and Eggs" was based on the distribution of values estimated in an LCA of butter across 21 countries [46]. Our land use estimate for beverages was lower than the estimated land use based on Agribalyse. Our water consumption estimate for meat analogs was somewhat lower than a previously reported estimate for 39 different meat analogs [48]. Although our mean GWP value for "Beverages" was somewhat higher due to the inclusion of alcoholic drinks, our mean GWP estimate for soda was within less than 0.1 kg $CO_2$-eq per kg of the value reported by Amienyo et al. in 2012 for carbonated soft drinks in glass bottles [52]. The wide variety of impacts associated with different beverages means that comparison of the category as a whole is troublesome, but our mean estimates are still reasonable when considering the impacts of various constituent beverages. Mixed dishes did not facilitate comparison to other sources due to their unique nature. For example, we did not find any LCA for most of the individual mixed dishes or a review containing them.

The main strength of this study was that, despite the effort and challenges involved in this LCA study, collecting this extensive dataset allowed us to assess a multiplicity of environmental impacts of a wide variety of foods, beyond the typical narrow focus on GHGEs. The land use and water consumption detailed in this paper were just an example of how the dataset allows a more comprehensive assessment.

The limitations of this study were primarily due to data availability. Our goal was to evaluate all AHS-2 FFQ foods from cradle to gate comprehensively, including all components of the lifecycle within those boundaries, and with data specific to the United States. However, despite gathering primary data and interpreting secondary data for this purpose, we had to rely on tertiary data for many of the foods and, further, had to exclude some foods due to lack of any data. Out of the original 278 foods we intended to assess, we completed an analysis of 248 foods. In addition, because the AHS-2 FFQ was designed specifically for the Adventist population in the USA and Canada, the foods contained in the questionnaire and, therefore, our analysis may not be representative of the general population. Finally, the age of the FFQ means that there are many new foods, especially meat analogs, which have come to market since it was created.

Future studies using this dataset will work to improve its comprehensiveness and accuracy. We plan to utilize the data collected as part of this research to provide further insights into connections between human and planetary health outcomes through the lens of food choices. One such exploration would include analyzing any synergies or tradeoffs involved when optimizing food choices for either health or minimal environmental burdens.

## 5. Conclusions

Previous work analyzing environmental impacts of foods has focused on a short list of products, mainly unprocessed ones, and singularly on GHGE. This study sought to expand the scope and enhance the detail of the assessment, as well as extend the range of environmental impacts considered. The resulting LCA of the AHS-2 FFQ foods evaluating their GWP, land use, and water consumption found results consistent with previously published sources. Plant-based foods had the overall lowest environmental impacts, while animal-based foods had the overall highest environmental impacts. This held true when comparing common sources of protein on a protein content functional unit basis. Finally, although ultra-processed foods had the highest GWP, processed foods had the highest land use, and unprocessed foods had the highest water use, implying that production of raw ingredients drives impacts more than their processing. Future work utilizing this dataset is planned to evaluate an array of environmental impacts of actual dietary choices among the AHS-2 questionnaire cohort. Ultimately, we strive to gain a better understanding of how the foods we eat can be selected in such a way to simultaneously maximize not only our own health but also that of our planet.

**Supplementary Materials:** The following are available online at http://www.mdpi.com/2071-1050/12/24/10267/s1, *Supplemental Data* (SD) Excel file with sheets, *Foods and System Boundaries*, *Environmental Impacts–Category*,

and *Environmental Impacts–Protein*; and *Supplementary Information* (SI) Word document, including Table S1: Energy Usage for Processing Foods, Table S2: Country of Origin for Out of Season Produce, and Table S3: International Transportation Distances by Mode.

**Author Contributions:** Conceptualization, A.B., U.F. and J.S.; Data curation, A.B. and U.F.; Formal analysis, A.B.; Investigation, A.B., K.J.-S., U.F., R.A.M. and A.M.; Methodology, A.B., U.F., A.C. and J.S.; Project administration, J.S.; Resources, K.J.-S., R.A.M., A.M., A.C. and J.S.; Supervision, A.B. and J.S.; Validation, A.B., A.C. and K.J.-S.; Visualization, A.B.; Writing—original draft, A.B. and J.S.; Writing—review and editing, A.B., U.F., A.C., K.J.-S. and J.S. Supplementary Materials–A.B., A.C., U.F., K.J.-S and J.S. All authors have read and agreed to the published version of the manuscript.

**Funding:** This research was funded by the McLean Fund for Nutrition Research, Loma Linda University.

**Acknowledgments:** The authors thank Andrew Mashchak for his initial assistance with data analysis, Emily Harima for gathering information regarding the transportation of out of season produce, and Laura Moore for her role in coordination and support.

**Conflicts of Interest:** The authors declare no conflict of interest. The funders had no role in the design of the study; in the collection, analyses, or interpretation of data; in the writing of the manuscript, or in the decision to publish the results.

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
