# Peer review of "Environmental Impacts of Foods in the Adventist Health Study-2 Dietary Questionnaire"

_sustainability, doi:10.3390/su122410267_

Round 1

Reviewer 1 Report

The paper aims at assessing the environmental impacts of the food items included in the food frequency questionnaire of the Adventist Health Study-2 survey. The problem stated is vital both for science and practice. I believe that the manuscript deserves for publication, however, it would be valuable to implement some improvements before the final acceptation. Below some specific comments are given.

Abstract

Novelty and value added of the study should be briefly described.

Keywords

It would be better not to repeat the words from the title of the paper as keywords.

Paper’s title

The title should be more directly linked to the aim of the paper – an environmental impacts should be clearly emphasized.

Introduction

Broader literature review should be presented in the introductory part of the paper or as a separate section. The most important achievements in the field studied should be briefly described and compared. It is not enough just to indicate them (line 52). Differences between the previous studies and the authors’ research should be presented, while the novelty and value added of the study should be comprehensively emphasised. Some aspects have been already inserted in the Discussion section, however, in my opinion these phrases fit better to the introductory part of the paper.

Materials and methods

Referring to Line 67: What are the other methods used for estimating environmental impacts of products? Why the LCA was chosen and what are its advantages over the others? Comparison between life cycle analysis and input-output analysis could be given to justify the method selection.

Line 75: What does it mean “a mixed methods LCA”? This concept should be clearly explained.

Line 183: It would be more precisely to use the name “the NOVA food classification system”.

The scope of the research should be determine more clearly: when the questionnaire survey was conducted? Was it nation-wide research? If not, the US states or Canadian counties/districts covered by the study should be mentioned. What was the socio-economic profile of the respondents? What about their dietary status?

Results and Discussion

Lines 231-234 should go before the research results described in the lines 224-227.

Line 263: “by” is missing before the bracket [22]

It would be valuable to employ cause-and-effects analysis to a broader extent and to present a bit more extensive discussion with previously published studies (the links to previous findings have been given very briefly in lines 315-319 only).

Limitations and directions for further studies (lines 25-334) fit better to the Conclusion section and they may be moved to this part of the paper.

Formal remarks

Titles of the Figures inside the Figure area are not necessary. It is enough to give them under the Figure. Units of the impacts assessed should be indicated in the titles or as notes  in figure footer. Sources of all Figures should be given as well.

Reviewer 2 Report

According to the reviewer, the presented work is not an article. It has a form of communication. The introduction is not a synthetic presentation of the research problem; it does not contribute much to work. In the introduction, literature is cited that refers to commonly known phenomena (e.g. lines: 34, 35, 39) while the discussion section is significantly limited.
The work has no novelty aspect. The discussion and conclusions contained in the conclusions section emphasize that the results obtained by the authors are consistent with the research of other authors.
It is well known that food of plant origin has a lower environmental impact than a food of animal origin. It is also evident that highly processed food should be eliminated from the human diet. Processed foods - limited. So how do the authors want to choose food in terms of human health, taking into account the consumption of processed food?
I propose in the future to separate the group: meat, poultry and fish, into meat, poultry and fish.
According to the reviewer, the indicated comments make it impossible to publish the article in this form.

Reviewer 3 Report

I find your paper very interesting. You address a current ad relevant problem using the correct methodology.  However, changes need to me made in other to make it a valuable contribution for this journal. Further effort in the literature review in the introduction and discussion section is needed.

Abstract

Implications of the study should be included.

Introduction

The introduction section is a bit simple and short. A more extensive literature review must be developed. A paper of this kind should include more that 26 references. A quick search on Scopus database including the words “LCA” and “food” results in up to 1,986 documents. I miss all that information in this section. On page 2 (lines 49 to 61) only 5 references are used to explain the previous work that has been done using LCA. Only in recent years dozens of papers about LCA have been published.

Materials and methods

A specific section about the database information should be included.

I am concerned about your sample. You have a big sample (96,000 people), but your sample only contains Adventist people. This reduces the chance to generalize your results to the general population and it is a clear limitation of your study that you should state clearly.

Do you have the data about the socioeconomic characteristics of your sample?

Section 2.1.1.2. System boundaries. I find all this information really useful. Please make sure that all the information is provided as this is crucial for the evaluation of your results and the comparison with other studies.

Page 3. Line 114. Please provide specific information about were have you get all the information. A table containing that information could be useful.

Line 146. Again, I am worried about all the assumptions. Please make sure you include all the information. Have you use any reference to make all these decisions?

Results

Figure 1. What foods are included in the Mixed dishes group?

All figures. Units are missed. Please ensure that all figures can be understood without the text.

Discussion

Again, I mis further comparison with other studies. Only reference to seven studies is included. This is not a proper discussion section, but a bit more elaborated results section. Please change all this section and make sure that you find studies to support your findings and ideas.

Limitations and information about future research should be included in the conclusions section.

Round 2

Reviewer 1 Report

The paper is much improved and I believe that the changes made have significantly enhanced the quality of the manuscript. However, I suggest two small improvements still to be made:

  1. Line 63-66: The socio-economic profile of the respondents and their dietary status should be described in more detail.
  2. Lines 436-476: Comparisons between the authors’ findings and results from previously published studies are added, however the description is of “technical” nature and some phrases are frequently repeated. If the authors consider it appropriate it would be worth to present all those similarities and differences in the table and give only a brief summarizing comment on it. Such a procedure would allow to avoid the problem of

Reviewer 2 Report

The authors' comments significantly improved the scientific value of the work.

Author Response

The reviewer's comment, "The authors' comments significantly improved the scientific value of the work," is appreciated. There does not appear to be any need for revisions on the basis of their comment.

Reviewer 3 Report

The upgrading in the Introduction section and the addition of supplementary data have improved the scientific soundness of your paper. 

Author Response

The reviewer's comment, "The upgrading in the Introduction section and the addition of supplementary data have improved the scientific soundness of your paper," is appreciated. There does not appear to be any need for revisions on the basis of their comment.